# Elastic–plastic fracture analysis of pressure pipelines with axial cracks based on the interaction integral method

Qi Song[1], Huifen Peng[2]*, Junyu Yao[2], Min Luo[2]

**1** Research Center of Coastal and Urban Geo-Technical Engineering, College of Civil Engineering and Architecture, Zhejiang University, Hangzhou, Zhejiang, China, **2** School of Mechanical Science and Engineering, Northeast Petroleum University, Daqing, Heilongjiang, China

* phfdaqing@163.com

**Data Availability Statement:** All relevant data are within the manuscript and its Supporting information files.

**Funding:** This research was funded by National Natural Science Foundation of China, grant number

## Abstract

The proposed work aims to demonstrate the significance of the plastic zone at the tip of an axial crack in a pipeline for managing Stress IntensityFactors(SIF). The three-dimensional finite element model of pressure pipeline with axial cracks was built by utilizing the Ramberg-Osgood X80 material model of pipeline. according to Von Mises yield criterion, the size of plastic zone at crack tip was determined, and the fracture parameters were calculated based on interaction integral method, the plastic stress deformation law, determination of elastic-plastic limit load and plastic correction of SIF at crack tip of pressure pipeline with axial crack were discussed. Consequently, it is observed that the elastic-plastic limit load diminishes as the initial crack length increases under specified pipe geometry and material conditions. the plastic zone dimensions at the crack tip of the pipeline expand proportionally with the relative crack length ($\delta$). Moreover, the relative error between the Stress Intensity Factors (SIF) before and after plastic correction exhibits nonlinear growth in response to increasing internal pressure within the pipeline. Notably, when assessing coefficients prior to plasticity corrections, it becomes evident that the maximum error may exceed 20% as the internal pressure rises. Importantly, the empirical verification data substantially aligns with the previously mentioned theoretical analysis results in a noteworthy concurrence.

## 1. Introduction

Across the globe, the prevailing trend in pressure pipeline development centers around achieving extended transmission distances, employing pipelines with larger diameters, and facilitating high-pressure applications [1, 2]. These advancements are primarily driven by continuous improvements in pressure transmission pipeline construction technology [3, 4]. However, it is worth noting that throughout the lifecycle of these pipelines, whether during their manufacturing, installation, or maintenance phases, the inevitable emergence of micro-cracks poses a significant challenge. Over time, these micro-cracks can escalate into critical issues, potentially culminating in catastrophic fracture accidents. Such unfortunate events not only jeopardize public safety but also result in substantial economic losses on a national scale. [5–10]

"51674088". The funders had no role in study design, data collection and analysis, decision to publish, or preparation of the manuscript.

**Competing interests:** The authors have declared that no competing interests exist.

The boundary element method was used [11] to effectively solve the boundary problem in the fracture analysis of cracked structures. ANSYS was used to evaluate the stress intensity factor of a high-pressure pipeline with multiple semi-elliptical cracks based on the virtual crack closure technique [5, 12, 13]. The above studies are based on linear elastic fracture theory and consider the effect of pipeline geometry, load, and crack parameters on the stress intensity factor without considering the effect of the plastic region at the crack tip on the fracture parameters of the member. As the stress at the crack tip of the pipeline tends to infinity in the order of $r^{-1/2}$, the effect of the plastic zone on the fracture parameters under the effect of infinite stress is not negligible [14–16]. The elastic–plastic confinement failure law of two-dimensional and three-dimensional cracks in X100 pipeline steel with perforated plate cracks based on the interaction integral method [17] was analysed to avoid the defect of crack tip meshing. The crack extension problem of X65 pipeline steel was examined [18, 19] using the interface unit CZM and compared it with experimental results to optimise the finite element model. The two-dimensional and three-dimensional crack constraint failure mechanism of axially straight cracked X100 pipeline steel was analysed [20–22] based on the numerical calculation method of elastic–plastic fracture mechanics. The coupled finite element meshless Galerkin approach was used to analyze the stress intensity factor of transverse surface elliptical cracks in carbon fiber wound repaired pressure pipes [23], which served as a useful guide for pipeline repair. The above elastic–plastic fracture analysis used the energy method and interface cells to examine the elastic–plastic restraint failure law and crack extension of pipelines with cracks [24–28] without determining the effect of the size of the plastic zone at the crack tip on fracture parameters. To accurately calculate the stress intensity factor and enhance the reliability of safety assessments for cracked pipelines, the study is structured as follows:Section 2 presents the theory of the interaction integral method, i.e. the plasticity correction stress intensity factor, is presented. The correction of the stress intensity factor is also based on the Ramberg-Osgood material model. In Section 3, a finite element model of a pressure pipeline with axial cracks is developed and experimentally verified.

In Section 4, we employ both the interaction integral method and the singular element method to investigate the influence of the plastic zone at the crack tip on the stress intensity factor.

Furthermore the effects of pipeline radius, thickness, initial crack length and internal pressure on the stress intensity factor are discussed.

Finally In section 5, the results obtained are summarized and a final conclusion is drawn. The novelty of this manuscript is mainly as follows:

1. The interaction integral technique is utilised to compute the stress intensity factor of pipeline fissures, thereby circumventing the reliance of conventional numerical fracture analysis on meshes, which in turn enhances the precision and efficiency of the calculations.

2. The text presents the Ramberg-Osgood model for pipeline steel material, analyzing the impact of minor plastic deformation near pipeline crack tips on the stress intensity factor, based on the elastic fracture mechanics of the Irwin line. This improves the dependability of pipeline safety analyses.

## 2 Materials and methods

### 2.1 Interaction integral method

Conventional finite element studies on crack extension rely heavily on the sparseness of the mesh, and a dense and distorted mesh reduces the computational accuracy and efficiency of

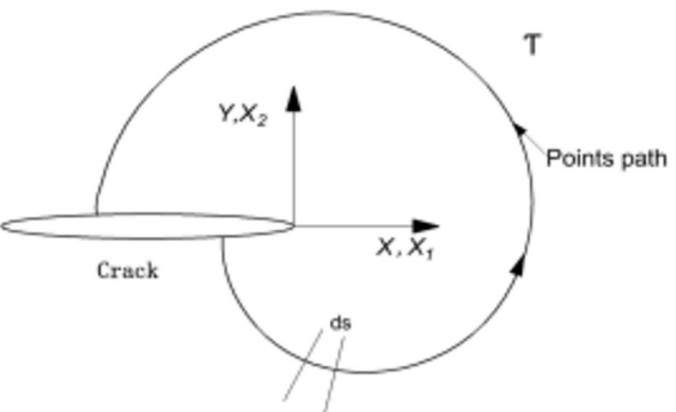

**Fig 1. Crack tip arbitrary integration loop Γ at crack tip.**

the finite element method. The interaction integral method can effectively avoid the limitations of conventional finite element numerical calculation of fracture parameters by establishing a local coordinate system at the crack tip and solving the fracture parameters using the interaction between the virtual and real fields, and this method is widely used in numerical analysis.

Fig 1 shows an arbitrary integration loop Γ around a crack tip whose energy integral can be defined as

$$J = \int_{T} \left( W dx_2 - T_i \frac{\partial u_i}{\partial x_1} dx_1 \right) \tag{1}$$

where the strain energy density $W = \int_0^{\varepsilon_{ij}} \sigma_{ij} d\varepsilon_{ij}$ and $\sigma_{ij}$ and $\varepsilon_{ij}$ are the stress and strain, respectively, $u_i$ is the component of the displacement vector, $T_i$ is the component of the force acting on the integral $T_i = \sigma_{ij} n_i$, and $n_i$ is the component of the unit vector outside the integral loop.

Establish real (1) and fictitious (2) stress-deformation fields, the latter being an arbitrary field satisfying the equilibrium and deformation coordination equations.

Superposition of the real and imaginary fields into Eq (1) can be performed based on Green's equation by applying the principle of superposition of linear elastic stress–strain as follows:

$$J_0 = \int_{\Gamma} \frac{1}{2} \left( \sigma_{jk}^{(1)} + \sigma_{jk}^{(2)} \right) \left( \varepsilon_{jk}^{(1)} + \varepsilon_{jk}^{(2)} \right) \delta_{1i} n_i ds - \int_{\Gamma} \left( \sigma_{jk}^{(1)} + \sigma_{jk}^{(2)} \right) \left( u_{j,1}^{(1)} + u_{j,1}^{(2)} \right) n_i ds = I + J^{(1)} + J^{(2)} \tag{2}$$

Similarly, $J_0$ for the joint action of real field (1) and imaginary field (2) can be expressed as follows:

$$J_0 = \frac{1}{E} \left[ \left( K_I^{(1)} + K_I^{(2)} \right)^2 + \left( K_{II}^{(1)} + K_{II}^{(2)} \right)^2 \right] = I + J^{(1)} + J^{(2)} \tag{3}$$

Performing a comparative analysis, we obtain

$$I = \frac{2}{E} \left[ K_I^{(1)} K_I^{(2)} + K_{II}^{(1)} K_{II}^{(2)} \right] \tag{4}$$

Solving for the parameter $K_I^{(1)}$, the imaginary field can be assumed to be a pure type I state and to satisfy:

$$K_I^{(2)} = 1, \quad K_{II}^{(2)} = 0 \tag{5}$$

This yields

$$K_I^{(1)} = \frac{E}{2} I \tag{6}$$

## 2.2 Correction of strength factors

When $r \to 0$, the crack tip yields to a small extent, and the stress near the crack tip tends to infinity in the order of $r^{-1/2}$; furthermore, the material is bound to yield under the action of the stress tending to infinity. Axial cracking of the pipeline is mainly a type I crack. Assuming that the pipeline is infinitely long and the diameter-to-thickness ratio is greater than 10, it can be regarded as a thin-walled pressure pipeline plane strain problem. To determine the yield size of the crack tip, the crack tip yield size is estimated based on the von Mises yield criterion; on the crack extension line ($\theta = 0$), the stress intensity factor $K_I$ is given by:

$$K_I = \beta_1 \sigma_\theta \sqrt{\pi a} = \beta_1 \frac{PR}{t} \sqrt{\pi a} \tag{7}$$

where $\beta_1$ is the geometric correction factor, $\sigma_\theta$ is the circumferential stress in the thin-walled pipeline, and $a$ is the initial crack length.

The stress field at the crack tip of a thin-walled pipeline under plane strain can be expressed as

$$\sigma_\theta = \frac{K_I}{\sqrt{2\pi r}}, \ \sigma_z = 2\mu \sigma_\theta = 2\mu \frac{K_I}{\sqrt{2\pi r}} \ \sigma_r = \tau_{r\theta} = \tau_{rz} = \tau_{z\theta} = 0 \tag{8}$$

where $\sigma_\theta$, $\sigma_z$, $\sigma_r$ are the circumferential, axial, and radial stresses of the pipeline, $\tau_{r\theta}$, $\tau_{rz}$, $\tau_{z\theta}$ are the shear stresses on the three faces of the pipeline, and $\mu$ is the Poisson's ratio of the thin-walled pipeline material.

According to the von Mises yield criterion, the pipeline crack tip yield size should be

$$r_p = \frac{(1 - \mu)^2}{\pi} \left( \frac{K_I}{\sigma_s} \right)^2 \tag{9}$$

where $\sigma_s$ is the yield strength of the pipeline.

For the Ramberg–Osgood material model,

$$\varepsilon = \frac{\sigma}{E} + \lambda \frac{\sigma_s}{E} \left( \frac{\sigma}{\sigma_s} \right)^n \tag{10}$$

where $\lambda$ is the material parameter, $n$ is the hardening index, and $E$ is the initial modulus of elasticity of the material.

Let the stress in the yield zone be $\sigma_s$ when the corresponding yield zone size is $r_p$; considering a small range of crack tip yielding, the strengthening curve AB can be replaced by the straight line A'B. At this time, the crack tip stress distribution is a straight line A'B with curve BK, as shown in Fig 2(a). Consider that the solution is required to meet the static equilibrium conditions after yielding; A'B above the shaded area provides greater yielding stress, increasing pipeline flexibility and forming more fractures. Therefore, the crack tip plastic area must be

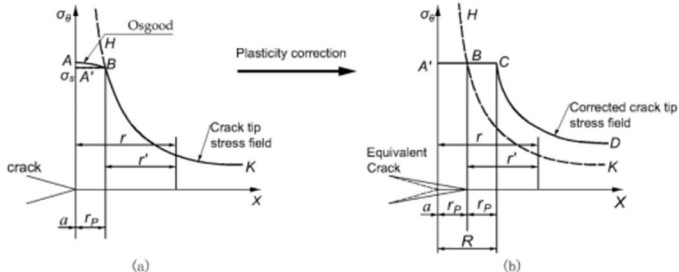

**Fig 2. Crack tip plasticity correction: (a) Crack tip plastic zone; (b) Corrected plastic zone.**

corrected. Assuming that the size of the modified yield zone is R, as shown in Fig 2(b), then the area under the solid line A'BCD in the cross-section.

## 3 Plastic stress intensity factor correction

The plastic yield zone at the crack tip results in an increase in the pipeline flexibility equivalent to an increase in the crack length. Hypothetically, if the crack size is considered to increase from $a$ to $+r_p$, then the above elastic solution is exactly CD. The term $a + r_p$ is known as the effective crack length. If $r' = r - r_p$, then the Irwin plasticity-corrected stress intensity factor is given as follows:

$$K_I' = \beta_1 \sigma_\theta \sqrt{\pi(a + r_p)} \tag{11}$$

On the crack extension ($\theta = 0$), the stress $\sigma_\theta$ is

$$\begin{cases} \sigma_\theta = \sigma_s & \text{when } r \leq R \\ \sigma_\theta = \dfrac{K_I}{\sqrt{2\pi(r - r_p)}} & \text{when } r \geq R \end{cases} \tag{12}$$

Substituting $r_p$ into Eq (11), the corrected stress intensity factor is

$$K_I' = \left[ 1 - \frac{1}{2}(1 - 2\mu)^2 \left( \frac{\sigma_\theta}{\sigma_s} \right)^2 \right]^{-\frac{1}{2}} K_I = \beta_2 K_I \tag{13}$$

where $\beta_2 = \left[ 1 - \frac{1}{2}(1 - 2\mu)^2 \left( \frac{\sigma_\theta}{\sigma_s} \right)^2 \right]^{-\frac{1}{2}}$ is the plasticity correction factor.

## 2.4 Grid sensitivity analysis

In order to analyze the sensitivity of the grid, setting node positions as shown in Fig 3, the results of the numerical analysis of the four nodes of the cracked tip with different mesh accuracies were analysed, as shown in Table 1, where the great relative error is defined as the relative error between the result of the calculation of the maximum number of cells and the number of nodes and the result of the calculation of the minimum number of cells and the number of nodes.

From the analysis results, it can be seen that the node VonMises stress near the crack tip of the pipeline gradually decreases with the increase of the distance from the crack tip, the same node with the increase of the number of cells and nodes, the error of the numerical calculation

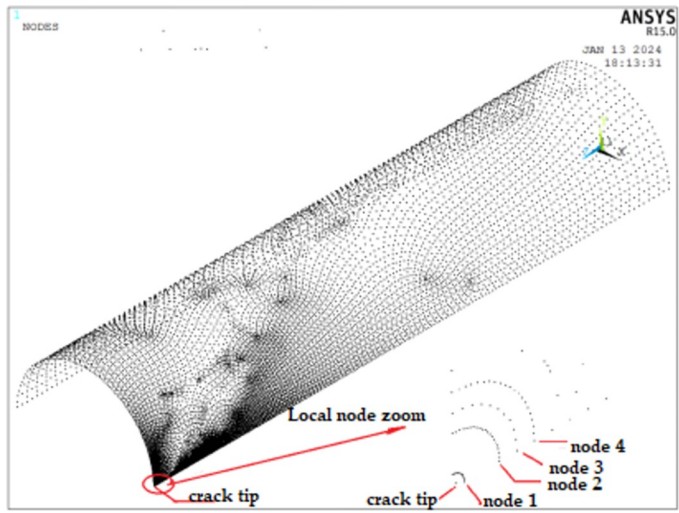

**Fig 3. Node number corresponding location map.**

results of the VonMises stress gradually decreases, and the relative error of the calculation results of different nodes with the minimum number of cells and nodes and the calculation results of the grid accuracy with the maximum number of cells and nodes is maximum 7.81% and minimum 5.10%, and it can be seen that the grid calculation accuracy of this paper meets the general accuracy requirements of the project. The relative error between the calculation results of different nodes with minimum number of cells and nodes and the calculation results with maximum number of cells and nodes is 7.81%, and the minimum is 5.10%, which can be seen from the above analysis.

## 3 Finite element model for pressure pipelines with axial cracks

### 3.1 Finite element model

Pipeline cracks are classified as surface, buried, and cracks; the first two types can be treated as cracks based on the method of equivalent force strength factor equivalence [29]. In this study, we focused on pressure pipelines with cracks, mainly type I cracks, Fig 4 shows detailed crack geometry in the model, with the following pipeline geometry: pipeline radius $R$ = 610 mm, pipeline wall thickness $t$ = 18 mm, pipeline length $l$ = 4000 mm, initial crack length $a$ = 15 mm; crack depth $h$ = 18 mm; material parameters: modulus of elasticity $E = 2.03 \times 10^{11}$, yield stress $\sigma_s$ = 560 MPa, Poisson's ratio $\mu$ = 0.25, Ramberg–Osgood stress–strain relationship $\varepsilon = \frac{\sigma}{E} + \lambda \frac{\sigma_s}{E} \left( \frac{\sigma}{\sigma_s} \right)^n$, hardening index $n$ = 13, material constant $\lambda$ = 1.07.

**Table 1. Grid sensitivity analysis.**

| Node number | VonMises stress/MPa | | | | | | Large relative error / % |
|---|---|---|---|---|---|---|---|
| | Number of units | 2227 | 2369 | 3453 | 4081 | 4786 | |
| | Number of nodes | 6680 | 7072 | 10384 | 12260 | 14367 | |
| Node 1 | | 229.59 | 235.59 | 242.96 | 246.89 | 247.52 | 7.81 |
| Node 2 | | 228.34 | 234.92 | 238.06 | 242.56 | 245.19 | 7.37 |
| Node 3 | | 227.79 | 231.85 | 235.06 | 238.66 | 241.45 | 5.10 |
| Node 4 | | 226.62 | 230.77 | 234.38 | 237.92 | 239.12 | 5.51 |

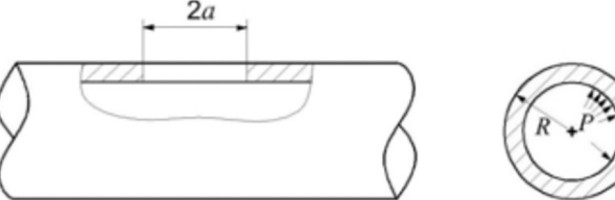

**Fig 4. Axial penetration cracks in pressure pipelines.**

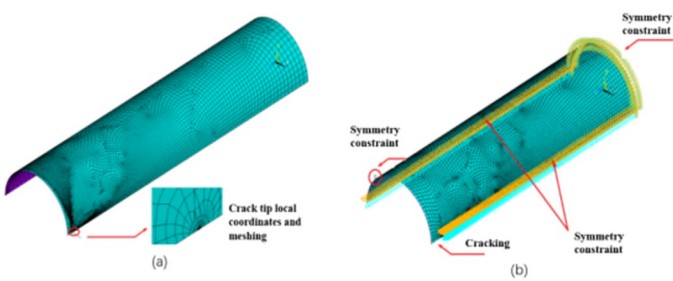

**Fig 5. (a) Finite element model of pipeline with axial cracks and (b) Pipeline restraint method.**

In this study, a three-dimensional elastic–plastic finite element model of a pressure pipelines with cracks was Developed using the self-compiling program APDL of ANSYS software [29, 30]. The analysis was conducted under the assumption of circumferential and axial symmetry throughout the entire pipeline, with the central crack serving as the base point. We selected 1/4 of the overall model as the calculation model and used the SHELL281 cell to divide the mesh for the SHELL281 isoperimetric singularity element. A local coordinate system was established at the crack tip, and the entire pipeline model cell was divided into eight layers along the thickness direction using the KSCON command of ANSYS software; Fig 5(a) shows the crack tip mesh model. The boundary conditions for the analysis are as follows: the right end face of the pipeline is fixed end constrained, the left end face and the tangent plane on the non-cracked side parallel to the crack plane are symmetrically constrained, and no constraint is added to the tangent plane on the cracked side parallel to the crack plane, as shown in Fig 5 (b). An elastic–plastic fracture analysis of the pressure pipelines with axial cracks was performed based on the aforementioned finite element model using the interaction integral method and the crack tip local coordinate system.

### 3.2 Experimental verification

While it is relatively straightforward to incorporate axial cracks in numerical simulations, conducting pressure pipeline tests with such cracks becomes challenging due to the potential for leaks. Consequently, for the purpose of verification testing, the specimens were specifically designed with non-penetrating axial cracks, eliminating the issue of leakage.

The specimen material was API X80 pipeline steel with an external diameter $D$ = 114mm, wall thickness $t$ = 4mm, initial crack length $2a$ = 30 mm and crack depth $h$ = 3mm. The circumferential strains at measurement points I and II (Fig 6) on the crack extension of the pipeline were obtained. The maximum pressure was set at 4 MPa to measure the circumferential strain values at the two measurement points at different pressures. Fig 7 shows the loading test setup.

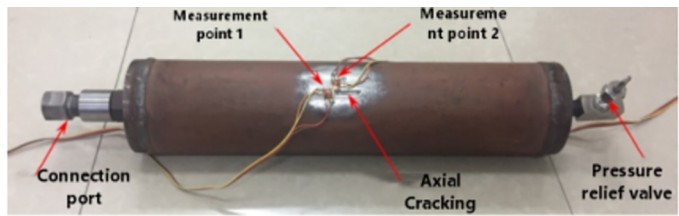

**Fig 6. Pipeline test points.**

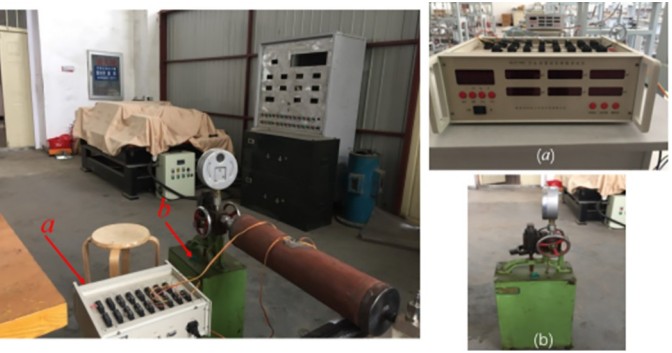

**Fig 7.** Test devices (a) XL2118C strain gauge. (b) S-SY-33 pressurized pump.

It can be observed from Table 2 that the maximum errors between the test and numerical simulation results for measurement points I and II occur at $P$ = 3 MPa with a maximum error of 9.00% and $P$ = 3 MPa with an error of 8.10%, respectively. The maximum error between the numerical simulation and test results is within the error tolerance, which indicates that the numerical model used in this study has high reliability.

## 4 Results and analysis

### 4.1 Effect of internal pressure on stress intensity factor

Table 3 lists the relative errors between the numerical solution of the stress intensity factor of the pipeline and results calculated using the empirical formula in the Handbook of Stress Intensity Factors for an initial crack length of the pipeline $a$ = 60 mm. The table data reveals

**Table 2. Comparison of annular strain test and numerical simulation results.**

| Measurement points | Calculation method | Internal pressure $P$ (MPa) | | | |
|---|---|---|---|---|---|
| | | **1** | **2** | **3** | **4** |
| Measurement points I | Numerical results ($\mu\varepsilon$) | 179.2 | 324.6 | 497.8 | 759.9 |
| | Test results ($\mu\varepsilon$)) | 167.5 | 301.9 | 456.7 | 699.7 |
| | Relative Error (%) | 6.98 | 7.52 | 9.00 | 8.60 |
| Measurement points II | Numerical results ($\mu\varepsilon$) | 240.3 | 513.6 | 740.7 | 1131.2 |
| | Test results ($\mu\varepsilon$) | 226.2 | 475.6 | 685.2 | 1047.6 |
| | Relative Error (%) | 6.23 | 7.99 | 8.10 | 7.98 |

**Table 3. Stress intensity factor relative error analysis table (*a* = 60 mm).**

| Stress intensity factor $K_I$ (N·mm$^{-3/2}$) | Internal pressure *P* (MPa) | | | | | |
|---|---|---|---|---|---|---|
| | 1 | 2 | 4 | 6 | 8 | 10 |
| Handbook of Stress Intensity Factors | 14.44 | 28.91 | 58.55 | 86.61 | 119.03 | 146.90 |
| Numerical solutions | 14.45 | 29.01 | 62.35 | 94.06 | 137.58 | 181.61 |
| Relative Error (%) | 0.09 | 0.37 | 3.24 | 8.60 | 15.58 | 23.63 |

that, when the internal pressure for internal pipeline pressures below 4 MPa, the relative error in stress intensity factor remains below 3.24%. This is primarily due to the fact that, at this pressure range, the crack tip either has not yielded or has just initiated yielding.

Under these conditions the effect of plastic deformation on the fracture parameters is minimal and plastic correction cannot be considered. When the internal pressure of the pipeline is greater than 4 MPa, the relative error of the stress intensity factor significantly increases with the increase in the internal pressure of the pipeline, and when the internal pressure of the pipeline is greater than 10 MPa, the relative error of the stress intensity factor has reached 23.63%. This indicates that the Stress Strength Factor Manual does not consider the effect of crack tip yielding on fracture parameters, which makes the safety assessment of high-pressure pipelines in service dangerous.

## 4.2 Pipeline crack tip elastoplastic analysis

The stress at the crack tip of the pipeline with axial penetration cracks yields, and the corresponding pressure is the elastic limit load $P_e$; when $K_I = K_{IC}$, the corresponding pressure is the plastic limit load $P_l$. Among them, $K_I$ is stress intensity factor of the pipeline with axial penetration cracks, $K_{IC}$ is plane strain fracture toughness of pipeline material. We used a three-dimensional elastic-plastic finite element model to obtain the plastic stress-strain change law in the area near the crack tip to determine the elastic-plastic limit load, the plastic deformation law of the crack tip, and the plastic influence region of the pipeline with axial cracks. According to von Mises yield criterion, the stress field at the crack tip is closely related to the elastic-plastic limit load and plastic strain.

Observing Fig 8, it can be seen that the von Mises stress at the crack tip is less than the yield strength at *P* = 2 MPa, reaches the yield stress at *P* = 4 MPa and the material begins to yield. When the internal pressure is 8MPa and 12MPa, the crack tip von Mises fluctuates around 560MPa and the mechanical response approximates an ideal elastic-plastic material model, with the stress near the crack tip increasing as the internal pressure increases, indicating that the plastic region of the crack tip is expanding.

Fig 9 shows the change in plastic strain with load sub-step for different internal pressures at the crack tip. It can be observed from the Fig 9 that the plastic deformation at the crack tip of the pipeline increases non-linearly with the increase in the internal pressure of the pipeline. The higher the internal pressure, the steeper the curve; when the internal pressure of the pipeline increased by a factor of 3 compared with the initial internal pressure, the maximum plastic strain at the crack tip of the pipeline increased from an initial 0.003 to 0.104, which is nearly 34.7 times the initial strain, indicating that the size of the plastic zone continues to expand under high pressure, resulting in the gradual weakening of the ability of elastic restraint plastic deformation and the gradual increase in pipeline flexibility. This indicates that more cracks are being created and therefore the stress intensity factor must be corrected.

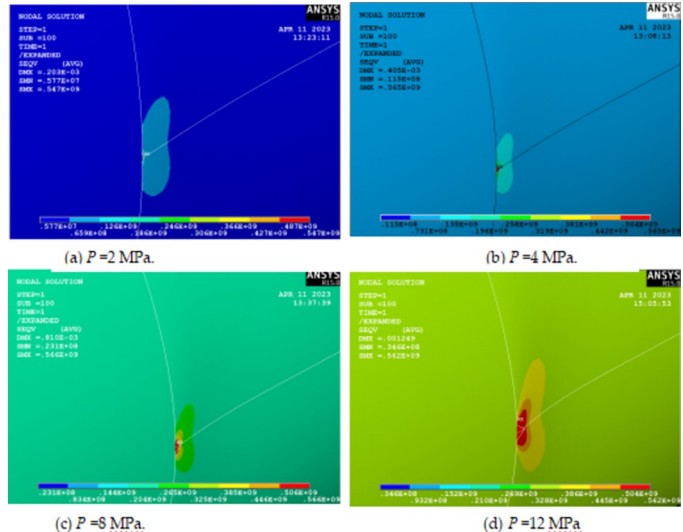

**Fig 8. Von Mises stress at crack tip under different pressures P.**

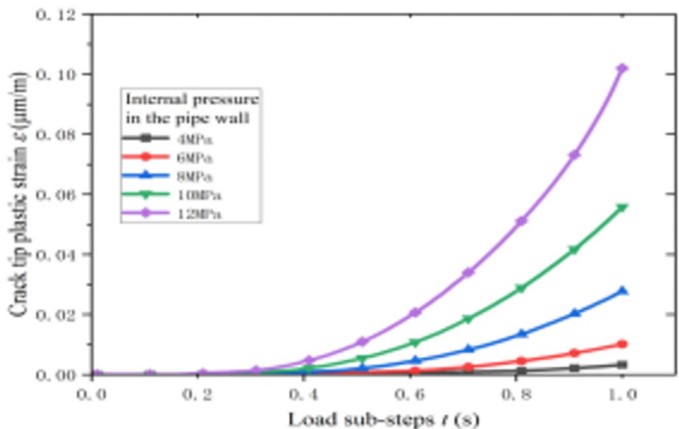

**Fig 9. Variations in the plastic deformation of the split tip with time at different internal pressures.**

Table 4 presents the relationship between the initial crack length, elastic–plastic limit load, and initial yielding time at $P$ = 18 MPa. It can be observed from the table that after determining the pipeline geometry and material parameters, the elastic–plastic limit load of the pipeline decreases and the time required for the initial yielding of the crack tip gradually decreases as the initial crack length of the pipeline increases.

**Table 4. Relationship between the initial crack length and elastic-plastic limit load.**

| Physical quantities | Initial crack length $a$ (mm) | | | | | | | | | | | |
|---|---|---|---|---|---|---|---|---|---|---|---|---|
| | 5 | 10 | 15 | 20 | 25 | 30 | 35 | 40 | 45 | 50 | 55 | 60 |
| Initial yielding time $t$ (s) | 19 | 13 | 11 | 9 | 6 | 6 | 5 | 5 | 5 | 4 | 4 | 4 |
| Elastic limit load $P_e$ (MPa) | 3.42 | 2.34 | 1.92 | 1.62 | 1.08 | 1.04 | 0.96 | 0.9 | 0.9 | 0.72 | 0.72 | 0.72 |
| Plastic limit load $P_l$ (MPa) | 17.6 | 16.6 | 13.7 | 14.9 | 13.9 | 13.0 | 12.5 | 12.1 | 11.2 | 11.4 | 10.9 | 10.4 |

**Table 5. Relative errors before and after $K_I$ plasticity correction.**

| $K_I$ N·mm$^{-3/2}$ | Initial crack length $a$ (mm) | | | | |
|---|---|---|---|---|---|
| | 20 | 30 | 40 | 50 | 60 |
| Before plasticity correction | 21.87 | 36.45 | 67.91 | 81.50 | 119.03 |
| After plasticity correction | 22.49 | 37.99 | 72.37 | 87.97 | 129.27 |
| Relative error (%) | 2.83 | 4.22 | 6.57 | 7.94 | 8.60 |

## 4.3 Impact factors of $K_I$ plasticity correction

Table 5 presents the relative error before and after plasticity correction of the stress intensity factor and initial crack length for pipelines with axial cracks at an internal pressure of 8 MPa. It can be observed from the table that the relative error before and after $K_I$ plasticity correction sharply increases in the non-linearity with the increase of the initial crack length; at $a = 60$ mm, which is thrice the initial crack length, the maximum relative error before and after $K_I$ plasticity correction is 8.60%. The fracture parameter before correction can be used in the assessment of the pipeline on the dangerous side.

Fig 10 shows the relationship between the plastic strain at the crack tip and the change in the load sub-step length for different initial crack lengths at $P = 12$ MPa. It can be observed from Fig 10 that the plastic deformation at the crack tip of the pipeline increases with the increase in the relative crack length. When the relative crack length $\delta \geq 0.007$, the plastic deformation at the crack tip increases sharply with the increase in the load sub-step length. When the relative crack length $\delta = 0.01$, the plastic strain at the crack tip is 0.26 and the corresponding plastic deformation is 10.4 mm. This indicates that the larger the relative crack length, the larger the plastic zone, the more flexible the pipeline, the longer the equivalent crack, and the higher the stress intensity factor. This suggests that as the relative crack length increases, so does the size of the plastic zone, resulting in increased pipeline flexibility, longer equivalent crack dimensions, and higher stress intensity factors.

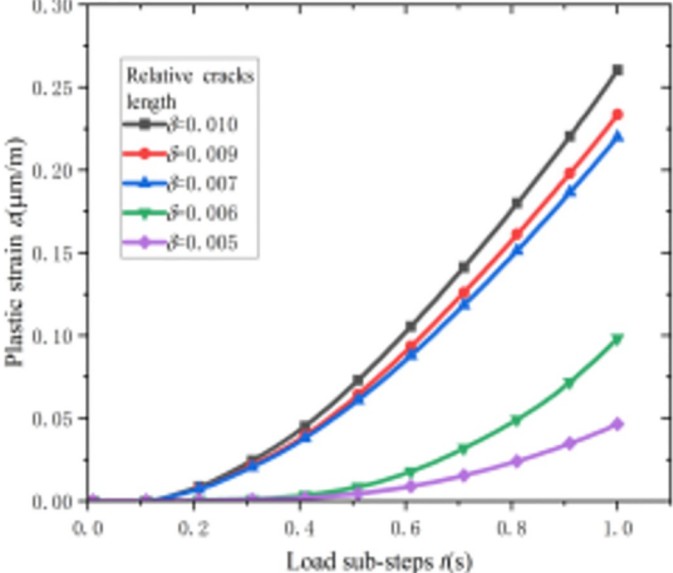

**Fig 10. Relationship between the crack tip plastic strain and load step length of the pipeline.**

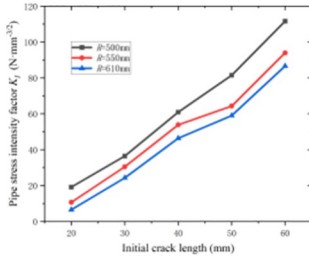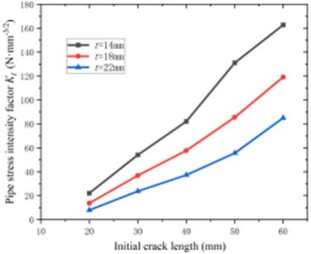

**Fig 11.** Variation law of stress intensity factor $K_I$ after plasticity correction: (a) Variations in $K_I$ with $a$ at different $R$. (b) Variations in $K_I$ with $a$ at different $t$.

Fig 11(a) shows the variation law of $K_I$ with the initial crack length $a$ for different pipeline radius $R$ when the pipeline thickness $t$ and pipeline length $l$ are certain. It can be observed from Fig 11(a) that $K_I$ increases slightly with the increase of the $R$ when the pipeline $t$ and $l$ are certain, and $K_I$ increases with the increase in the initial crack length for the same pipeline radius $R$.

Fig 11(b) shows the variation law of $K_I$ with the initial crack length $a$ for different pipelines $t$ when the pipeline $R$ and $l$ are certain. It can be observed from Fig 11(b) that $K_I$ decreases with the increase in the pipeline thickness $t$ when the pipeline $R$ and $l$ are certain, and increases with the increase in the initial crack length for the same pipeline $t$.

In summary, the method proposed in this paper can effectively support pipeline safety assessments, particularly when the ratio of initial crack size to the plastic zone size at the crack tip complies with conditions of limited yielding. To further enhance the comprehensiveness of this research field, future studies could expand their scope to encompass fracture analysis in pipelines with extensive yielding at the crack tip, thereby establishing more efficient models that consider both large-scale yielding at the crack tip and the incorporation of the Dugdale model.

## 5 Conclusion

This paper introduces a three-dimensional finite element model for conducting elastoplastic fracture analysis in pipelines containing axial cracks. We propose a novel fracture analysis method based on the Ramberg-Osgood material model and the Mises yield criterion, which accounts for the plastic zone effect at the crack tip. This method takes into consideration the initial crack length, pipeline geometric parameters, and internal pressure.

Our approach provides accurate calculations of the stress intensity factor and is capable of predicting the pipeline's elastic and plastic limit pressures. To quantify the influence of the plastic zone at the crack tip on the pipeline's stress intensity factor, we employ the relative error before and after the plastic correction of the stress intensity factor. The effectiveness of the proposed method is demonstrated through experimental results.

Several key findings are listed below:

1. The SIF increases as the pipeline radius increases, and it decreases as the pipeline thickness increases.

2. The elastic and plastic limit load of the pipeline decreases as the initial crack length increases, and the time required for the initial yield decreases gradually as the initial crack length increases.

3. The change rule of the plastic zone at the crack tip with pipeline relative crack length is non-linear. When the pipeline relative crack length is smaller than 0.007, the nonlinear change is not obvious, when the pipeline relative crack length reached 0.01, plastic zone at the crack tip increases sharply as the relative crack length increase, which caused a further decrease in Elastic constraint of pipeline, indicating that the plastic zone at the crack tip expanded unlimitedly, which leads to the failure of the pipeline.

4. The relative error of the SIF before and after plasticity correction overall increased as the internal pressure of pipeline increases. the relative error increased relatively slowly when the internal pressure was in the range of 1 MPa and 4 MPa and then increased quickly when the internal pressure was bigger than 6 MPa, the highest relative error is 23.63% when the internal pressure was equal to 10 MPa, which means it is necessary to consider the effect of crack tip plasticity on the SIF When the internal pressure of the pipeline is higher than 8 MPa.

In summary, the method proposed in this paper can effectively support pipeline safety assessments, particularly when the ratio of initial crack size to the plastic zone size at the crack tip complies with conditions of limited yielding. To further enhance the comprehensiveness of this research field, future studies could expand their scope to encompass fracture analysis in pipelines with extensive yielding at the crack tip, thereby establishing more efficient models that consider both large-scale yielding at the crack tip and the incorporation of the Dugdale model.

## Supporting information

**S1 Data. In accordance with Fig 10.**
(XLSX)

**S1 File. APDL program in accordance with Fig 8 and Table 2.**
(DOCX)

**S2 File. APDL program in accordance with Fig 11 and Tables 3 and 5.**
(DOCX)

**S3 File. APDL program in accordance with Table 4.**
(DOCX)

**S4 File. APDL program in accordance with Fig 5.**
(DOCX)

## Author Contributions

**Conceptualization:** Qi Song, Huifen Peng, Min Luo.

**Data curation:** Qi Song.

**Formal analysis:** Huifen Peng.

**Funding acquisition:** Min Luo.

**Investigation:** Min Luo.

**Methodology:** Qi Song.

**Project administration:** Huifen Peng, Min Luo.

**Resources:** Min Luo.

**Software:** Qi Song.

**Supervision:** Qi Song, Huifen Peng.

**Validation:** Huifen Peng, Min Luo.

**Visualization:** Qi Song.

**Writing – original draft:** Huifen Peng.

**Writing – review & editing:** Qi Song, Junyu Yao.

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
