## [Decision Letter · Decision Letter 0]

30 Aug 2023

PONE-D-23-24376Elastic–plastic Fracture Analysis of Pressure Pipelines with Axial Cracks Based on the Interaction Integral MethodPLOS ONE

Dear Dr. Peng,

Thank you for submitting your manuscript to PLOS ONE. After careful consideration, we feel that it has merit but does not fully meet PLOS ONE’s publication criteria as it currently stands. Therefore, we invite you to submit a revised version of the manuscript that addresses the points raised during the review process.

ACADEMIC EDITOR:I just received comments from one reviewer.Both English and presentations should be significantly improved. The novelty of this manuscript should be clearly stated through literature review.Refertences with last three years should be updated to help you well define your research problem. ==============================

We look forward to receiving your revised manuscript.

Kind regards,

Jianguo Wang, PhD

Academic Editor

PLOS ONE

Journal Requirements:

3.We note that the grant information you provided in the ‘Funding Information’ and ‘Financial Disclosure’ sections do not match. 

Please include your amended statements within your cover letter; we will change the online submission form on your behalf."

Reviewers' comments:

Reviewer's Responses to Questions

**Comments to the Author**

1. Is the manuscript technically sound, and do the data support the conclusions?

Reviewer #1: Partly

2. Has the statistical analysis been performed appropriately and rigorously? 

Reviewer #1: Yes

3. Have the authors made all data underlying the findings in their manuscript fully available?

Reviewer #1: Yes

4. Is the manuscript presented in an intelligible fashion and written in standard English?

Reviewer #1: No

5. Review Comments to the Author

Reviewer #1: The proposed work aims to prove the important role of the plastic zone at the crack tip of a pipeline with axial crack to deal with the stress intensity factor. A review of the full paper found that acceptance in its current form for publication in the journal of PLOS ONE is not recommended. The paper is poor-written. Reconsider after addressing the following problems:

1. English style is the primary problem. Currently, I cannot understand this study clearly unless a native speaker to re-check the grammar and typos.

2. There are too much old literature cited in the reference list, should replace which with some recent literatures about soil research and application of soil uncertainties. Authors can consult: https://doi.org/10.1016/j.tws.2020.107404; https://doi.org/10.1016/j.psep.2020.10.005; https://doi.org/10.3390/s22030986.

3.Discussions need to increase the broad applicability of the model. I cannot capture the novelty of this work.

4.Conclusion part is poor written, the core contributions are required to list.

6. PLOS authors have the option to publish the peer review history of their article (what does this mean?). If published, this will include your full peer review and any attached files.

Reviewer #1: No

---

## [Author Response · Author response to Decision Letter 0]

26 Oct 2023

On behalf of my co-authors, we are very grateful to you for giving us an opportunity to revise our manuscript. we appreciate you very much for your positive and constructive comments and suggestions on our manuscript. We have studied reviewers' comments carefully and tried our best to revise our manuscript according to the comments. The following are the responses and revisions I have made in response to the reviewers' questions and suggestions on an item-by-item basis. In this revised version, changes to our manuscript were all highlighted within the document by using red-colored text. Thanks again to the hard work of the editor and reviewer!

---

## [Decision Letter · Decision Letter 1]

1 Dec 2023

PONE-D-23-24376R1Elastic–plastic fracture analysis of pressure pipelines with axial cracks based on the interaction integral methodPLOS ONE

Dear Dr. Peng,

Thank you for submitting your manuscript to PLOS ONE. After careful consideration, we feel that it has merit but does not fully meet PLOS ONE’s publication criteria as it currently stands. Therefore, we invite you to submit a revised version of the manuscript that addresses the points raised during the review process.

We look forward to receiving your revised manuscript.

Kind regards,

Jianguo Wang, PhD

Academic Editor

PLOS ONE

Reviewers' comments:

Reviewer's Responses to Questions

**Comments to the Author**

1. If the authors have adequately addressed your comments raised in a previous round of review and you feel that this manuscript is now acceptable for publication, you may indicate that here to bypass the “Comments to the Author” section, enter your conflict of interest statement in the “Confidential to Editor” section, and submit your "Accept" recommendation.

Reviewer #1: (No Response)

2. Is the manuscript technically sound, and do the data support the conclusions?

Reviewer #1: Partly

3. Has the statistical analysis been performed appropriately and rigorously? 

Reviewer #1: N/A

4. Have the authors made all data underlying the findings in their manuscript fully available?

Reviewer #1: Yes

5. Is the manuscript presented in an intelligible fashion and written in standard English?

Reviewer #1: No

6. Review Comments to the Author

Reviewer #1: 1. It is not yet evident in detail how the selection of the boundary condition for the constraint of the fixed end of the right end face of the pipeline is based on the actual physical conditions of the pressure pipeline. Suggest providing a reasonable explanation.

2. The article did not analyze the sensitivity of the grid, which may cause some confusion. Although the interaction integration method can improve the accuracy and efficiency of calculating fracture parameters and reduce the requirements for grids, it does not mean that the influence of grids does not need to be considered entirely. Therefore, it is recommended that relevant explanations be provided for the lack of grid sensitivity analysis.

3. It is recommended to provide clearer crack modeling in the finite element model diagram so that readers can understand the model more clearly. This may include more detailed crack geometry in the model.

7. PLOS authors have the option to publish the peer review history of their article (what does this mean?). If published, this will include your full peer review and any attached files.

Reviewer #1: No

---

## [Author Response · Author response to Decision Letter 1]

1 Feb 2024

we hereby confirm that this manuscript is our original work and has not 

been published previously nor has it been submitted simultaneously elsewhere, no conflict of interest exists in the submission of this manuscript, and the manuscript is approved by all authors for publication.

---

## [Decision Letter · Decision Letter 2]

11 Mar 2024

Elastic–plastic fracture analysis of pressure pipelines with axial cracks based on the interaction integral method

PONE-D-23-24376R2

Dear Dr. Peng,

We’re pleased to inform you that your manuscript has been judged scientifically suitable for publication and will be formally accepted for publication once it meets all outstanding technical requirements.

Kind regards,

Jianguo Wang, PhD

Academic Editor

PLOS ONE

Additional Editor Comments (optional):

Please provide a proofreading version for your final submission. Some attentions should be paid to the consistency of terminology in the whole manuscript.

Reviewers' comments:

Reviewer's Responses to Questions

**Comments to the Author**

1. If the authors have adequately addressed your comments raised in a previous round of review and you feel that this manuscript is now acceptable for publication, you may indicate that here to bypass the “Comments to the Author” section, enter your conflict of interest statement in the “Confidential to Editor” section, and submit your "Accept" recommendation.

Reviewer #1: All comments have been addressed

2. Is the manuscript technically sound, and do the data support the conclusions?

Reviewer #1: Yes

3. Has the statistical analysis been performed appropriately and rigorously? 

Reviewer #1: Yes

4. Have the authors made all data underlying the findings in their manuscript fully available?

Reviewer #1: Yes

5. Is the manuscript presented in an intelligible fashion and written in standard English?

Reviewer #1: Yes

6. Review Comments to the Author

Reviewer #1: All my concerns have been addressed. I would like to recommend accepting this paper for publishing on PLOS ONE.

7. PLOS authors have the option to publish the peer review history of their article (what does this mean?). If published, this will include your full peer review and any attached files.

Reviewer #1: No

---

## [Editor Report · Acceptance letter]

15 Mar 2024

PONE-D-23-24376R2 

PLOS ONE

Dear Dr. Peng, 

I'm pleased to inform you that your manuscript has been deemed suitable for publication in PLOS ONE. Congratulations! Your manuscript is now being handed over to our production team.

Kind regards, 

on behalf of

Dr. Jianguo Wang 

Academic Editor

PLOS ONE